# Changes of Ex Vivo Cervical Epithelial Cells Due to Electroporation with JMY

**DOI:** 10.3390/ijms242316863

**Published:** 2023-11-28

**Authors:** Henriett Halász, Zoltán Szatmári, Krisztina Kovács, Miklós Koppán, Szilárd Papp, Edina Szabó-Meleg, Dávid Szatmári

**Affiliations:** 1Department of Biophysics, Medical School, University of Pécs, 7624 Pécs, Hungary; henriett.halasz@aok.pte.hu (H.H.); edina.meleg@aok.pte.hu (E.S.-M.); 2Kazincbarcika Hospital, 3700 Kazincbarcika, Hungary; szatzol@yahoo.com; 3Department of Pathology, Medical School, University of Pécs, 7624 Pécs, Hungary; kovacs.krisztina2@pte.hu; 4DaVinci Clinics, 7635 Pécs, Hungary; mkoppan@gmail.com (M.K.); veloxetferox@gmail.com (S.P.)

**Keywords:** actin, JMY, ex vivo epithelial cells, cytopathology, actin-binding proteins

## Abstract

The ionic environment within the nucleoplasm might diverge from the conditions found in the cytoplasm, potentially playing a role in the cellular stress response. As a result, it is conceivable that interactions of nuclear actin and actin-binding proteins (ABPs) with apoptosis factors may differ in the nucleoplasm and cytoplasm. The primary intracellular stress response is Ca^2+^ influx. The junctional mediating and regulating Y protein (JMY) is an actin-binding protein and has the capability to interact with the apoptosis factor p53 in a Ca^2+^-dependent manner, forming complexes that play a regulatory role in cytoskeletal remodelling and motility. JMY’s presence is observed in both the cytoplasm and nucleoplasm. Here, we show that ex vivo ectocervical squamous cells subjected to electroporation with JMY protein exhibited varying morphological alterations. Specifically, the highly differentiated superficial and intermediate cells displayed reduced nuclear size. In inflamed samples, nuclear enlargement and simultaneous cytoplasmic reduction were observable and showed signs of apoptotic processes. In contrast, the less differentiated parabasal and metaplastic cells showed increased cytoplasmic activity and the formation of membrane protrusions. Surprisingly, in severe inflammation, vaginosis or ASC-US (Atypical Squamous Cells of Undetermined Significance), JMY appears to influence only the nuclear and perinuclear irregularities of differentiated cells, and cytoplasmic abnormalities still existed after the electroporation. Our observations can provide an appropriate basis for the exploration of the relationship between cytopathologically relevant morphological changes of epithelial cells and the function of ABPs. This is particularly important since ABPs are considered potential diagnostic and therapeutic biomarkers for both cancers and chronic inflammation.

## 1. Introduction

The key step of the stress response is the entry of Ca^2+^ into the cell, which is triggered by the activity of protein kinases [1,2]. Ca^2+^ signalling controls a wide range of cellular functions and, more recently, research has revealed that an increase in Ca^2+^ within the nucleoplasm governs processes that are distinct from those influenced by cytosolic Ca^2+^ [3,4,5,6]. This suggests the possibility of distinct roles for Ca^2+^ in the nucleus and the cytoplasm. The Ca^2+^-sensitive regulators of the cytoskeleton can rearrange the entire network and therefore actin-binding proteins (ABPs) are implicated in the morphological signs of stressed cells in various cytopathological cases [7,8,9]. The main functions of the cytoskeletal machinery of the eukaryotic cell are cell shape determination, cell motility, division, and contraction. Furthermore, they play an important role in intracellular transport and signalisation [10], as well as in intercellular connections. The precise mechanism by which the Ca^2+^ signal initiates the signalling of apoptotic factors is less elucidated [11]. Furthermore, it is unclear what sensory pathway distinguishes between the processes of apoptosis and differentiation and how these processes impact the reorganisation of cytoskeletal components. Investigations are also required to explore how malfunctioning apoptotic processes or signalling may influence the remodelling of cytoskeletal elements during tumorigenesis. A wide range of processes associated with inflammation (e.g., cytokine and interleukin stimulated pathways, high concentration of intracellular Ca^2+^) reconfigure the entire machinery of the cellular stress response system [12]. Abnormally expressed actin-binding proteins (ABPs) are involved in chronic inflammation and the progression of cancers [8]. The involvement of actin in the nuclear complexes controlling, e.g., chromatin remodelling and histone modifications, was previously reported. Nevertheless, it is crucial to explore future research directions to provide further insights into the functional importance of these processes [9]. The great diversity of stress response processes associated with Ca^2+^ and actin in the cytoplasm [1,13,14], distinct from those in the nucleus and mitochondria [4,5,6], presents an intriguing aspect for exploring their potential sensory function in cell differentiation or apoptosis.

In our research, in addition to other ABPs, we are interested in the function of the junctional mediating and regulating Y protein (JMY) [15,16] as an important but understudied ABP. JMY can be transported into the nucleus where it enhances the transcription of the proapoptotic p53 [17] and other p53-dependent genes [18]. The p53 and ABP, as well as their complexes, are regulatory elements in the cytoskeletal remodelling and motility both in the cytoplasm and the nucleoplasm [19], and, thus, play an important role in tumorigenesis and invasion [20,21,22,23,24,25,26,27,28,29,30]. JMY is a neuronal Wiskott–Aldrich syndrome protein (N-WASp) homologous Arp2/3 helper protein that induces the differentiation of HL-60 cells into highly motile neutrophil-like cells [19]. JMY binds to the pointed end of actin filaments and helps the network-building machinery of Arp2/3 complexes [15,17,19,31,32]. However, in the cytoplasm, JMY and Arp2/3 complexes can arrest the E-cadherin exhibition, thus leading to a high number of extracellular interactions and quick migration of metastatic cells [33]. JMY is prominently expressed in lymph nodes, primary colorectal, and head–neck carcinoma cells, but its expression is diminished in breast carcinoma cells [34]. The effect of JMY on differentiated cells is less studied. Our prior model [16], based on in vitro data, indicates that the p53-JMY complex formation and their function are Ca^2+^-dependent and modifies the division and motility of HeLa cells. The p53-JMY cotransport into the nucleus enhances p53 expression [19]. JMY in a complex with actin monomer and p53 accelerates the division of stressed HeLa cells by an unknown pathway [16]. We hypothesised that a reduced level of JMY-p53 complex in the nucleus may hinder apoptosis, while an elevated intracellular level of the complex could trigger it due to the increased p53 expression [16].

The exploration of novel biomarkers for the improvement of cytopathology directs our interest in the ABPs, which can be important as a linkage between apoptotic processes and morphological transformations. Here, we are studying the possible relationship between morphological signs of ex vivo ectocervical squamous cells and the function of intracellularly exceeding JMY.

The process of inflammation is a physiological reaction with the response of a high number of inflammatory cells. The morphological alterations of epithelial cells are combined with the appearance of one or more well-identified pathogenic agents, leukocytes, and an exudate background. Morphological alterations linked to elevated cytokine levels are indicative of inflammatory atypia. For instance, in squamous cells, one can observe nuclei with irregular contours, condensed chromatin (pyknosis) and subsequent fragmentation (karyorrhexis). Nuclei may also be enlarged with clear, finely granular chromatin, the cytoplasm is frequently vacuolated and sometimes clear and small perinuclear halos are also visible [35,36].

ASC-US (Atypical Squamous Cells of Undetermined Significance) means the finding of abnormal cells in the tissue that lines the outer part of the cervix. ASC-US is the most common abnormal finding in a Papanicolaou (PAP) test. The ASC-US represents 4–6% of cytological changes and is quantitatively or qualitatively insufficient for a definitive interpretation. It can be a sign of infection with certain types of human papillomavirus (HPV) or other types of pathogens, such as fungal infection. However, well-defined criteria for ASC-US are not established. The features may range from a marginal increase in the size of the nucleus to nuclear hyperchromasia and irregular nuclear membranes [11]. Atypical parakeratosis and repair are included in the ASC-US category. Essentially, these changes fall outside the range of normal cytological limits. ASC-US sometimes appears with definite nuclear enlargement or with other nuclear abnormalities (binucleosis, nuclear enlargement and chromatin irregularities) [35,36]. The inflammation rules out the usage of the ASC-US category and, thus, we need to investigate the differences between inflammation and ASC-US.

Bacterial vaginosis is a common disease with moderate discomfort and vaginal discharge, rarely with pruritus that is linked to the presence of Gardnerella vaginalis, which causes severe atypical morphological signs of the cells at the microscopic level [37].

In this study, we explore cytoskeletal processes related to JMY in the context of cytopathology. We focus on the significant morphological changes resulting from increased intracellular JMY levels in ex vivo epithelial cells under in vitro conditions. Specifically, we examined cell number, nucleus, and cell size, as well as any specific morphological change attributable to a response to electroporation with JMY of ectocervical squamous cells. Cells were obtained from routine cervix screening. Surprisingly, cells responded to an excess of intracellular JMY levels with altered cell survival and various morphological changes.

## 2. Results

This study provides data that were distinguished by Bethesda System (TBS) categories, using common cytomorphological patterns for clear identification [37,38] of the selected ectocervical squamous cells [39].

We carried out electroporation with JMY assays and square-wave pulses were used to introduce JMY in human ectocervical squamous cells to exceed its intracellular level. A low-voltage electroporation system was applied that we previously developed and successfully used in our study [16]. Experiments were carried out on cytology samples from 20 women with 52.4 ± 16.3 years of average and 55 years of median ages, from the range of 20–73 year-old patients. Taking into account the age and hormonal differences of the patients, two groups were created: pre-menopausal (7 women) and post-menopausal (13 women). Based on the analysis of cell types with PAP-staining from liquid-based preparations, samples were categorised as normal (15 women) or inflamed cases (4 women). All inflamed samples contained leukocytes and showed morphological signs of bacterial infection (e.g., clue cells, bacterial cell debris in the background). One of the inflammatory samples was signed as severe bacterial vaginosis. Based on the macroscopic observations and the official cytology reports of samples (validated by Dept. of Pathology, Kazincbarcika Hospital, Hungary), five specimens showed signs of atrophy, 3 of cervical ectropium and 2 of hyperaemia; furthermore, two specimens were afflicted by atrophy and ectropium along with mild inflammatory symptoms. One sample showed the morphological signs of ASC-US. Control samples were chosen randomly, with one sample selected from pre-menopausal cases and another from post-menopausal cases, both of which were electroporated using a buffer-only solution. Samples were obtained on the same day and all assays were carried out simultaneously. It is not possible to select control samples for all relevant categories prior to PAP-staining. Electroporation and subsequent recovery require 24 h, because unexpected alterations and intracellular processes can be initiated in ex vivo epithelial cells that have been exposed to in vitro conditions for more than two days [40,41,42]. We were unable to select control samples after the initial PAP-staining. Given all the constraints, within this study, our primary focus lies in examining the combined effect of in vitro incubation, JMY, and electroporation.

In normal samples, the average number of intermediate and superficial cells, e.g., cell density in a unit ROI (region of interest) area reduced from 47 ± 22.4 to 31.9 ± 17.3 after the electroporation with JMY, and we observed a confluent saturation (Figure 1A,B,E). In inflamed samples, the electroporation with JMY caused similar effects. The average cell number was also decreased (from 30.38 ± 10.3 to 16.38 ± 9.6); additionally, dead cells underwent a significant transformation, resulting in a remarkable amount of cell debris and a high number of particles (Figure 1C,D,E). Basically, the inflammation directly reduced the number of cells from 47 ± 22.4 to 30.38 ± 10.3, but due to the electroporation in the absence of JMY, it was increased from 47 ± 22.4 to 61.73 ± 24.5 (Figure 1E) and resulted in a definitely high difference between the samples of normal cells electroporated in the absence and presence of JMY, 31.9 ± 17.3 and 61.73 ± 24.5, respectively. The JMY treatment reduced the cell density from 52.3 ± 27.3 to 43 ± 26.4 in the pre-menopausal group and from 44.4 ± 19 to 26.7 ± 15.1 in the post-menopausal group (Figure 1F).

For the interpretation of the JMY effect on the cell and the nuclear size, we defined the average diameter of cells or nuclei. Interestingly, the size of the normal cells did not change after the electroporation with JMY (50.4 ± 9.7 μm and 50.5 ± 9 μm, non-electroporated and electroporated, respectively) (Figure 2A,B,D,E and Figure 4A). In the absence of JMY due to the electroporation against buffer only, the size of the normal cells showed a minor increment (5%) from 50.4 ± 9.7 μm to 53 ± 8.7 μm (with equal median values) (Figure 2A–C and Figure 4A). Cells became full of small, dense volumes, making their cytoplasm rougher due to electroporation than in non-electroporated cases. The electroporation with JMY reduced (by 16%) the average size of nuclei from 8.7 ± 2 μm to 7.29 ± 1.6 μm (Figure 2A,B,D,E and Figure 4B). In the absence of JMY, the size of nuclei did not show a remarkable difference 8.7 ± 2 μm and 8.86 ± 1 μm due to the electroporation (Figure 2A–C and Figure 4B). The average values of the calculated nuclear/cell size ratio were between 15% and 18% in each case and showed the same tendency as nuclear size differences (Figure 4C).

In inflamed cases, the electroporation with JMY caused a significant shrinkage of the cells; their diameter was decreased (by 8%) from 47.89 ± 10.8 μm to 44 ± 8.4 μm (Figure 3A–D and Figure 4A), and their nuclei were enlarged from 8.53 ± 1.5 μm to 9.4 ± 1.8 μm (10% increase) (Figure 3A–D and Figure 4B). In addition, a high number of dense particles appeared. In inflamed cases, the electroporation with JMY, compared to normal cells, increased from 17% to 21% only (Figure 4C).

Regarding normal cells, the average cell size of pre- and post-menopausal samples was identical and independent of the electroporation with JMY (Figure 5A). However, cells from post-menopausal samples showed some nuclear enlargement, 9.2 ± 2.1 μm vs. 7.47 ± 1.6 μm (in pre-menopausal samples). The JMY electroporation reduced the nuclear size only in post-menopausal samples to 7 ± 1.6 μm and did not change it in pre-menopausal cases 7.4 ± 1.6 μm (Figure 5B). The average nuclear/cell size ratio values were between 14% and 20% in each case and showed the same tendency as nuclear size differences (Figure 5C).

Interestingly, parabasal and metaplastic cells (Figure 6A,B) were lysed by electroporation against buffer only in the absence of JMY (Figure 6C). In the presence of JMY, cells responded to electroporation with filopodia and membrane tube formation [43] in pre-menopausal samples (Figure 6D). However, extensive cytoplasmic protrusion formation was observable in post-menopausal samples, which showed signs of atrophy and mild inflammation (Figure 6E).

The sample of the severe bacterial vaginosis contained a high number of clue cells with asymmetric, binucleated and hyperchromatic nuclei with a perinuclear halo (Figure 7A,C). JMY treatment reduced the nuclear irregularities but did not modify the cytoplasmic abnormalities (cytoplasmic clearings with sharp borders) (Figure 7B,D).

In the sample that showed characteristics of ASC-US (Figure 8A–D), JMY treatment reduced the nuclear and perinuclear irregularities but cytoplasmic clearings still remained (Figure 8B,D,).

## 3. Discussion

Prior studies analysed the relationship between cytological details of clinically relevant disorders and the actin content of squamous cells [44,45,46]. Here we tried to make a pioneer study in molecular cytopathology, since this is the first time that we are exploring the possible relationship between cytopathologically relevant morphological changes of ex vivo ectocervical squamous cells and the function of an intracellularly exceeding ABP, which is considered candidate for diagnostic and therapeutic biomarker. Consequently, we need to figure out the possible pathways, which are implicated in the process of JMY-modified cytoplamsmic and nucleoplasmic reconstruction. Interpreting our data, we created a hypothetical model (Figure 9) to predict these possible pathways, which can work as an outlook for further studies.

Our observations point out the possibility that differentiated cells can be stimulated by the simultaneous effect of in vitro incubation and electrical pulses since their population size was increased, but it was decreased in the presence of JMY. Interestingly, the cell number from pre-menopausal cases appears to be less sensitive to the elevated JMY levels. This suggests that estrogen may have a protective effect on cells against the influence of JMY. Additionally, we can assume that JMY is involved in the elimination of certain types of cells in post-menopausal cases. As a consequence of electrical pulses and in vitro incubation, it is possible for all cells to experience an increased intracellular Ca^2+^ concentration as part of their primary intracellular stress response, from pCa8 to pCa4 (where pCa = −log [Ca^2+^]) [47]. This can serve as an effective stress response and lead to the alteration of the functional hierarchy of ABPs [48]. The Ca^2+^ concentration can regulate the rearrangement of the actin network, in addition to others, the cell motility and division due to the Ca^2+^-sensitive feature of JMY [16]. The large variety of receptors plays an important role in the rearrangement of actin filaments due to the activation of RhoA, cdc42, Rac and PI3K [49]. The cellular activity of Rho, cdc42, Rac and PI3K pathways is controlled by hormonal changes and depends on the stage of differentiation [50,51,52,53,54,55]. Estrogen and estradiol mainly activate the PI3K cascade and increase cell survival [56], promote differentiation [57] and protect the cells from apoptosis [58,59].

We assume that the low cell density after the JMY treatment can be attributed to the activation of Rho-ROCK, which leads to changes in the rearrangement of tight-junctions and alterations in the contractility and motility of the perijunctional actin network [14,60,61]. However, the estrogen-sensitive contribution of Rho and Rac pathways [54,62] can be implicated in post-menopausal cells and causes their relatively high distribution. In normal epithelial cells, the Arp2/3 helper function of JMY seems to be the most important to interpret the specific effects due to electroporation with JMY [63]. The presence of pyknotic, smaller nuclear size of superficial and intermediate cells after the electroporation with JMY is a crucial morphological change by the treatment. Cells from post-menopausal cases show more distinct differences than pre-menopausal ones, which were harmonised with the nuclear size of pre-menopausal samples due to JMY treatment. We can suggest that estrogen protects cells against significant nuclear enlargement. All pyknotic nuclei of the JMY-treated cells may be indicative of chromatin condensation [64,65] and actin translocation. Both can be induced by RhoGTPase glycosylation [66] and by Rho-mDia pathway with ABPs as formin, profilin [49,67,68,69] and different WASp homologous proteins [70]. Finally, the initiation of actin nucleation in the cytoplasm and the subsequent stabilisation of filaments, along with the export of monomers to the nucleus, resulted in a reduction in the nuclear size. The high Ca^2+^ concentration in stressed cells may directly activate the pathway of Cdc42 cascade [47] and caspase-catalysed proteolysis [71], thus, can play an important role in the formation of cytoplasmic clearings. Electroporation with JMY can stimulate cytoskeletal activity in less differentiated parabasal and metaplastic cells by cdc42 and Arp2/3 [72]. The activation of these actin-reassembling pathways can induce filopodia and membrane tube formation [73] in cooperation with the PI3K-Akt pathway [74]. Interestingly, membrane tubes frequently form between stressed and neighbouring unstressed cells and help cell survival under stress conditions [75,76].

Prior to the sampling, the inflammation process resulted in high levels of cytokines, interleukins [77]. Inflammatory processes can reduce the survival of stressed cells [78]; therefore, after the sampling, we observed a relatively small population of cells. However, subsequently, the remaining level of cytokines and interleukins can induce intracellular activation of the p53 and RhoA, Rac pathways [61,79,80]. Then, permanently high intracellular Ca^2+^ can induce p53 dimerisation and activation [16,81]. As was observed previously, cytokine treatment induced the formation of long, dense F actin-based stress fibres and the actin concentration, which was significantly increased compared to untreated cells [82]. Here, the remaining intermediate and superficial cells in the inflammatory samples after the JMY treatment showed nuclear enlargement, as it was observed previously [19]. High intracellular Ca^2+^ with JMY-p53 complexes and the activated RhoA-mDia profilin pathway [49] can lead to shrinkage of cytoplasm and swelling of nuclei, which is possibly driven by the translocation of actin into the nucleus. Simultaneously, in the same sample, the presence of a high number of small dense particles may allow us to suggest that the remarkable loss of cells is the consequence of the apoptosis of stressed cells and possibly the dense particles correspond to apoptotic bodies. Cells undergo dramatic morphological changes to form apoptotic bodies and need a cytoskeletal reorganisation [83,84]. The enhanced level of p53 and JMY can result in the activation of apoptosis [85], as we expected in our previous model [16]. In parabasal and metaplastic cells of atrophic samples with mild inflammation, simply the stress of the electrical pulse in the absence of JMY caused immediate cytoplasmic lysis and nude nuclei, which we can explain by the activation of cdc42 linked caspase cascade [47,71,86]. Nevertheless, in the presence of JMY, cells formed cytoplasmic protrusions, implying that JMY enhances the motility and cytoskeletal activity of the less differentiated cells from inflammatory samples. We obtained similar findings by studying HeLa cells’ motility and division by scratching assays [16]. If we try to interpret molecular processes in their background, we can assume that the high JMY content due to the interleukins can trigger the Rac-WAVE-Arp2/3 pathway [87] and stimulate cytoplasmic protrusion formation [80,88]. However, the inflammation accelerated mTORC1/2 promoting Arp2/3 complex function [89] and different ABPs can cause frequent branching and a meshwork of actin filaments [54,90].

Hyperchromatic cells with nuclear and cytoplasmic irregularities in case of severe bacterial vaginosis of a pre-menopausal woman raises the possibility that the interleukins modified the contribution between PI3K and p53 Rho-Rac [54,55,56,62,91], therefore elucidating the presence of binucleated superficial and intermediate cells by irregular nuclear division (Rho, cdc42) [92,93] and condensed chromatin (Rac) [87,94]. The cytoplasmic clearing can be linked to the cdc42-dependent caspase-induced proteolysis [71]. The JMY possibly in a complex with p53 diminished nuclear and perinuclear irregularities and asymmetries possibly by resting of the Rho-mDIA pathway. Furthermore, the activity of cdc42 and PI3K pathways seemed to be independent of JMY treatment and did not modify the cytoplasmic abnormalities.

The abnormal appearance of epithelial cells in ASC-US shows high morphological similarity to cells affected by vaginosis, suggesting that the background molecular processes of ASC-US and vaginosis are overlapping. In the absence of cytokines and interleukins, the estrogen stimulates the PI3K pathway [56] and the contribution of Rho and Rac pathways [54,62] can be disturbed by an unknown signal and cause nuclear and perinuclear irregularities by the activation of cdc42 [92,93] and Rac [87,94]. The cytoplasmic clearing with a sharp border can be linked to the cdc42-dependent caspase cascade [71]. Identically, with the sample from the vaginosis and ASC-US cases, the JMY-p53 complex possibly diminished nuclear and perinuclear irregularities and did not recover the cytoplasmic abnormalities. The role of actin filament dynamics in controlling apoptosis signalling is clearly demonstrated by the fact that tumour cells have evolved mechanisms to promote apoptotic processes. It was anticipated that actin and pathways associated with actin-binding proteins in apoptosis are essential [95].

In summary, we can assume that JMY can alter nuclear and perinuclear morphology through the Rho-ROCK-mDia pathway, which may also be related to apoptotic processes. Furthermore, JMY can drive cytoplasmic rearrangement via the cdc42-Rac-PI3K and WASP-WAVE pathways.

## 4. Materials and Methods

### 4.1. Proteins

The mouse JMY (pET20b construct, from Prof. Dyche Mullins Lab, UCSF, San Francisco, CA, USA) was expressed in *E. coli* BL21 Rosetta cells, induced by 1 mM IPTG and purified with Ni-NTA affinity chromatography under imidazole buffer conditions. This method was described previously [96]. It was then stored in Ca^2+^-free buffer A (0.2 mM ATP, 2 mM TRIS, pH 7.4). 

### 4.2. Ex Vivo Epithelial Cells

We obtained ectocervical squamous cells due to routine cervical screening (Regional Research Ethical Committee of Clinical Centre at University of Pécs (7531-PTE2019)), when after the examination the gynecologist put the used cytology brush into 5 mL Leibovitz medium (Sigma-Aldrich, St. Louis, MO, USA). Due to the subsequent incubation (16 h on ice) all remaining cells were sedimented, and then we harvested living cells by 2000× *g* for 5 min centrifugation. Pelleted cells were divided into two volumes, both were in 5 mL of fresh Leibovitz medium, one for low-voltage electroporation with JMY [16], and one for immediate staining with PAP Red stain kit (Abcam, Cambridge, UK). PAP-staining was carried out applying the manufacturer’s protocol, as fixation with alcohol, then in successive steps, staining with hematoxylin, OG-6 then EA-50. Twenty-four hours following the low-voltage electroporation, samples were prepared by PAP-staining. Referring to the Bethesda System (TBS) categories for reporting cervical cytology definitions, we stated the specific signs of normal, as well as inflammatory samples, and also the cases of severe bacterial vaginosis and ASC-US [35,36]. Experiments were carried out on cytology samples from 20 women with 52.4 ± 16.3 years of average and 55 years of median ages, from the range of 20–73 year-old patients.

### 4.3. Low-Voltage Electroporation to Increase the Intracellular Level of JMY

We developed a low-voltage electroporation system [16], which, based on an ArduinoUno^®^ board, leads 5 V, 0.46 mA current in 1 s ON/6 s OFF square-wave pulses, each administered 92 μF in 5 V/cm between two electrodes for 10 min on cell culture to introduce the JMY protein inside of the cells. Typically, epithelial cells can be transfected with 250- 900 μF in a single pulse of 200–300 V/0.4 cm (http://biorad-ads.com/transfection_protocols/ (accessed on 15 November 2023)), which results in 50–270 mC of charge; in our assay, only 39 mC was administered. Cells were in 5 mL of Leibovitz medium completed with 125 nM of JMY placed in a sterile Falcon tube; the majority of cells were in the suspension. and a few of them were sedimented at the bottom. In a previous step, we made 2 holes on the lids to put in one-one steel electrode and fixed them with silicone glue, electrodes were pieces of a stainless steel wire (ø 0.2 mm). We provided 1 cm distance between them, subsequently connected to the outputs of the board, and the 5 V/cm electrical field was obtained between the electrodes only. The output current was measured directly at the end of the electrodes. The cells were electroporated in a sterile environment.

### 4.4. Microscopy

After the fixation and labelling of cells, we examined their morphology with an Olympus IX 81 inverted light microscope (Olympus Europa Gmbh, Hamburg, Germany) with 10× and 40× objective lenses.

### 4.5. Image Analysis

ImageJ (developed by NIH, Bethesda, MA, USA) was applied for the quantitative analysis of microscopy images. The average number of intermediate and superficial cells in a unit ROI (region of interest) area was analysed at 5 certain points of the image in the area of 500 μm × 500 μm; then, cells were counted manually. For the interpretation of the cell and nuclear size we determined their average. First, we measured the average diameter of cells or nuclei on the mean in five independent directions per cell then calculated the average of them in the population of 20–50 cells from the same sample.

### 4.6. Statistics 

The statistical analysis was performed in Origin 2018 (OriginLab Corp. Northampton, MA, USA), which was derived from independent samples. The effects of JMY treatment were studied on the population of cells and presented with box diagram. Our aim was to obtain and measure the effect of treatment on the population of cells. Therefore, two sample *t*-tests were used; then, to avoid the possibility of the data variance-based Type I error, we tested their significance by ANOVA and Bonferroni tests (Appendix A). Statistically significant differences between groups were defined as *p* values < 0.05 and are indicated in the legends of the figures.

## 5. Conclusions

Ex vivo, ectocervical squamous cells react to an excess intracellular JMY level with altered cell survival and various morphological changes. In vitro incubation and low-voltage electrical pulses stimulated superficial and intermediate epithelial cells since their number increased, but in the presence of JMY, the population of treated cells became smaller. Differentiated superficial and intermediate cells exhibited decreased nuclear size, whereas cytoplasmic activity and protrusions were observed in less differentiated parabasal and metaplastic cells. Morphological signs such as responses were altered by inflammation or hormonal effects. Estrogen could protect cells from the effects of JMY treatment. Inflammatory conditions also seem to have a strong effect on the cytoskeleton. Thus, the survival rate of squamous cells was reduced, and their cytoplasm shrunk, whereas their nuclei were enlarged due to the high cytoplasmic level of JMY. In addition, inflammatory cells in severe vaginosis showed irregularities similar to those in ASC-US. Surprisingly, in both cases, JMY can only affect the nuclear and perinuclear irregularities of the differentiated cells, which we can describe by the possible overlaps between the pathways in the background of vaginosis and ASC-US. In addition, we can assume that the free cytoplasmic concentration of p53 and Ca^2+^ play an important role in the reconstruction of nuclear and cytoplasmic morphological differences between cells from inflamed or normal cases, whether p53 can form Ca^2+^ dependent complexes with JMY or not. 

Our observations can provide an appropriate basis for the exploration of an expected sensory system that links the cytopathologically relevant morphological changes (binucleosis, metaplasia, nuclear enlargement and chromatin irregularities, etc.) of epithelial cells to the function of JMY, which is a considered candidate diagnostic and therapeutic biomarker for cancers and chronic inflammation.

## Figures and Tables

**Figure 1 ijms-24-16863-f001:**
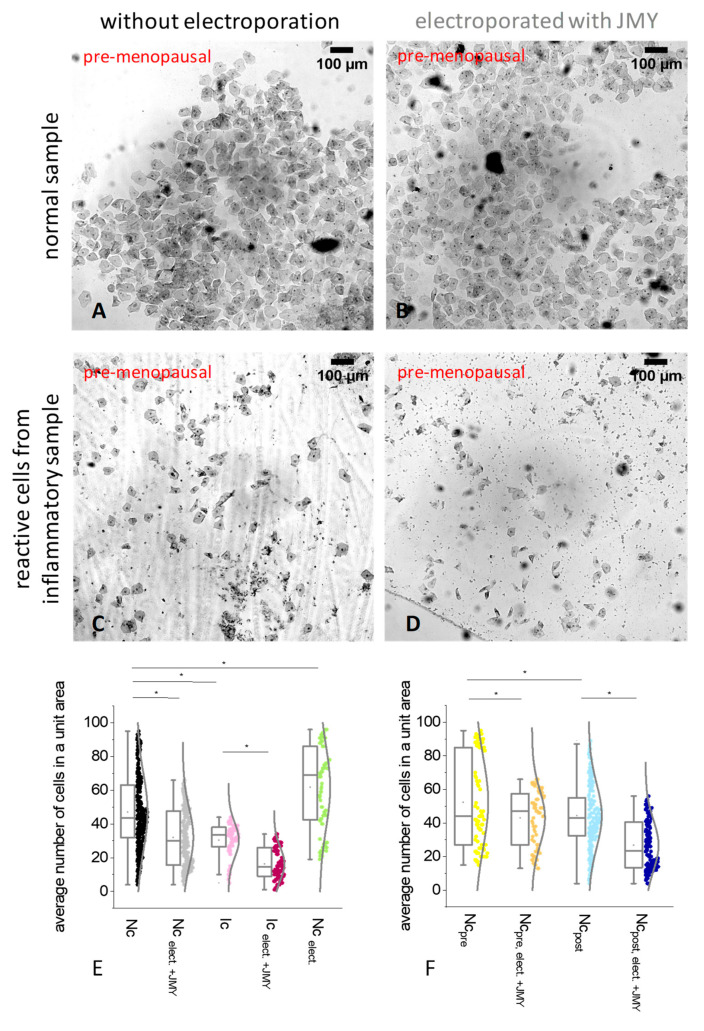
Microscopy imaging and analysis of PAP-stained ectocervical squamous cells in which cells were either non-electroporated or electroporated with JMY. (**A**) Non-electroporated intermediate and superficial cells from normal samples show less confluent saturation than (**B**) cells electroporated with JMY. (**C**) Non-electroporated inflammatory intermediate and superficial cells show similar confluent saturation than (**D**) inflammatory intermediate and superficial cells electroporated with JMY (colours of the titles on panels (**A**–**D**) indicate the colours of groups on panels (**E**,**F**). (**E**,**F**) Statistical analysis of the cell density in 5 certain points of the images in the area of 500 μm × 500 μm. Box normal diagrams represent data with ±SD (error bars), average value (points), and median values (lines). (**E**) Intermediate and superficial, non-electroporated cells from normal samples (Nc, black circles, *n* = 240), or electroporated with JMY (Nc _elect_._+JMY_, gray circles, *n* = 218), or electroporated in the absence of JMY (Nc _elect_., green circles, *n* = 120), inflammatory intermediate and superficial, non-electroporated cells (Ic, pink circles, *n* = 104), or electroporated in the presence of JMY (Ic _elect_._+JMY_, purple circles, *n* = 99). (**F**) Patients were classified into two groups, 6 pre-menopausal women and 13 post-menopausal women. Non-electroporated normal intermediate and superficial cells from pre-menopausal women (Nc _pre_, yellow circles, *n* = 80), or electroporated with JMY (Nc _pre_. _elect_._+JMY_, orange circles, *n* = 80). Non-electroporated normal intermediate and superficial cells from post-menopausal women (Nc _post_, blue circles, *n* = 160), or electroporated with JMY (Nc _post_. _elect_._+JMY_, dark blue circles, *n* = 168). Asterisk means statistically significant differences between groups. Significances were defined as *p* values < 0.05.

**Figure 2 ijms-24-16863-f002:**
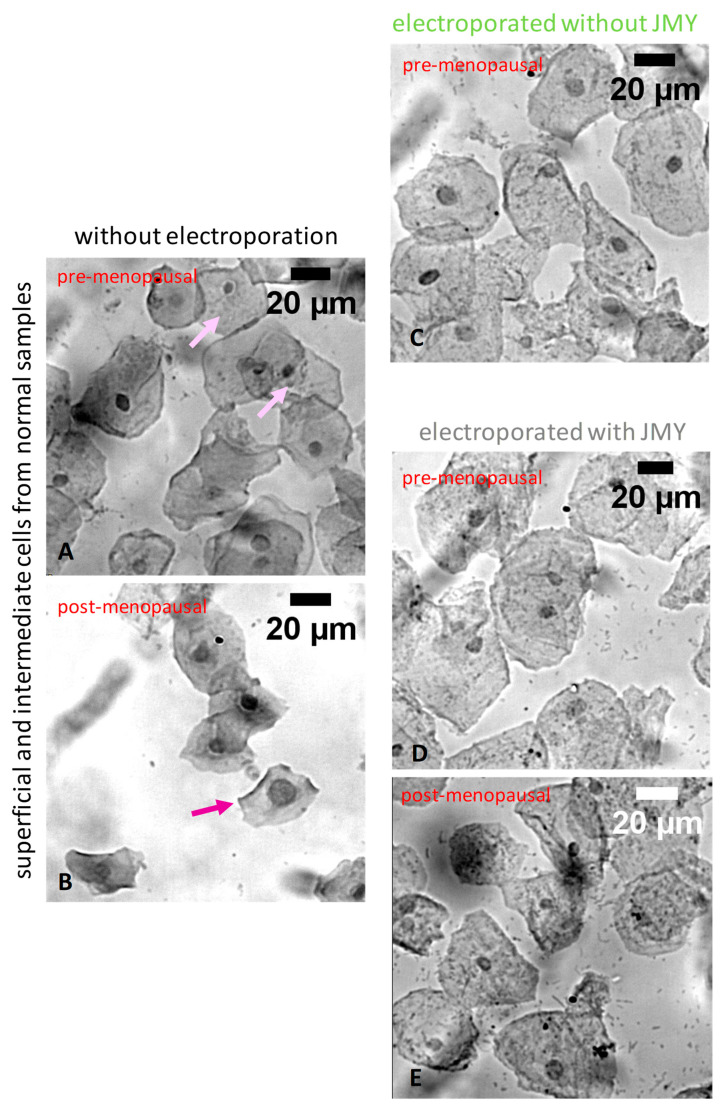
Morphological differences of PAP-stained ectocervical squamous cells, in which cells were non-electroporated or electroporated with JMY. (**A**,**B**) Typical intermediate (purple arrow) and superficial cells (pink arrows) in non-electroporated samples. (**C**) Intermediate and superficial cells after the electroporation against buffer only, in the absence of JMY. (**D**,**E**) Pyknotic cells after the electroporation with JMY (colours of the titles on the panels indicate the colours of groups in Figure 4).

**Figure 3 ijms-24-16863-f003:**
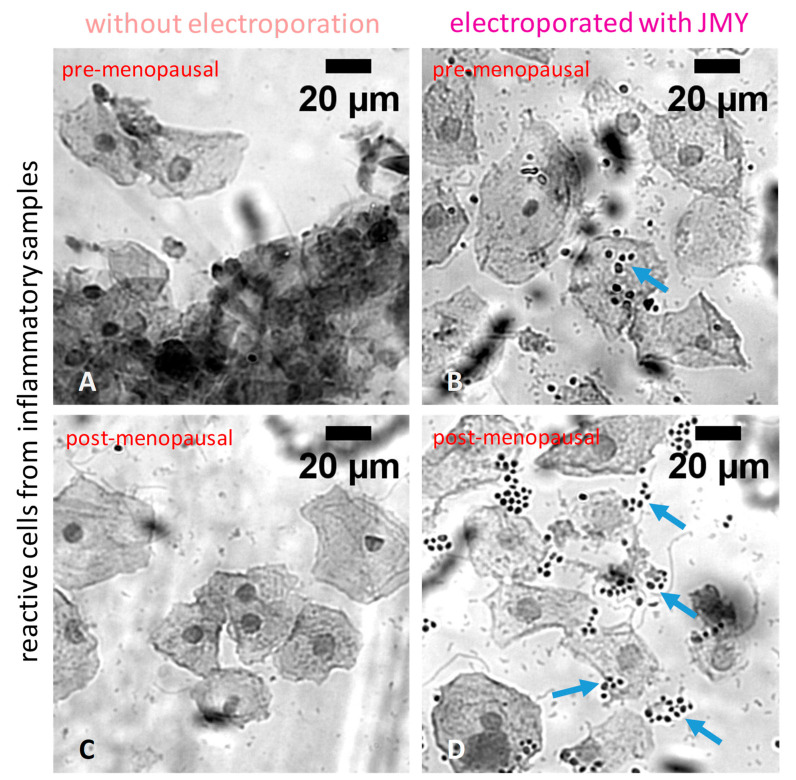
Morphological differences of PAP-stained reactive ectocervical squamous cells from inflammatory samples in which cells were non-electroporated or electroporated with JMY. (**A**,**C**) Non-electroporated inflammatory intermediate and superficial cells (**B**,**D**) or inflammatory intermediate and superficial cells electroporated with JMY. Electroporation resulted in the occurrence of enclosed hyperchromatic bodies (blue arrows) (colours of the titles on the panels indicate the colours of groups in Figure 4).

**Figure 4 ijms-24-16863-f004:**
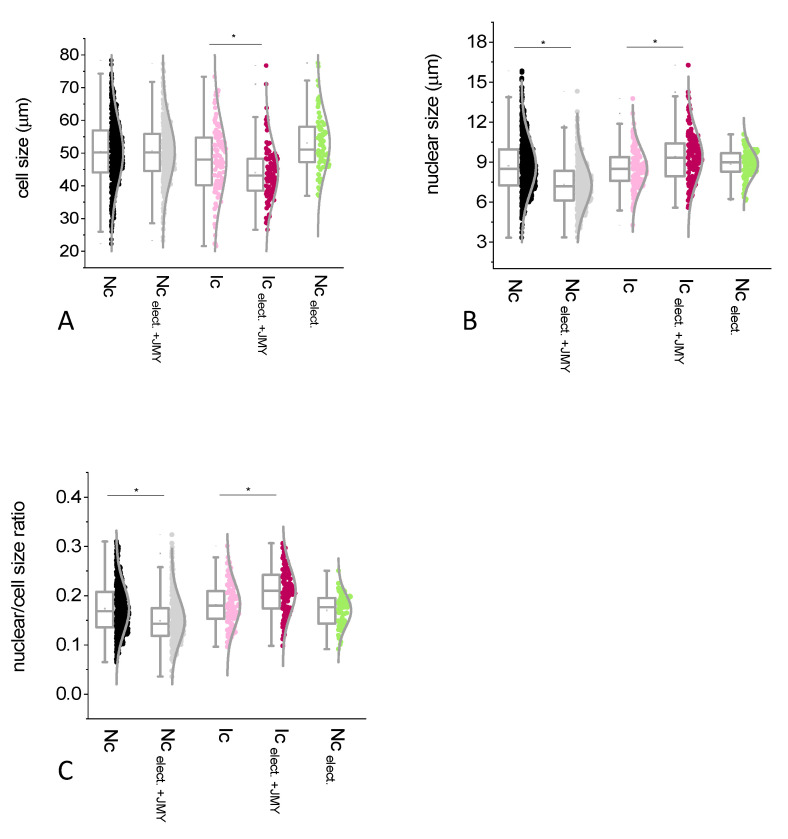
Analysis of the size differences of ectocervical squamous cells, in which cells were non-electroporated or electroporated with JMY. Box normal diagrams represent data with ±SD (error bars), average value (points), and median values (lines). We studied (**A**) the cell size, (**B**) the nuclear size, then calculated (**C**) the nuclear and cell size ratio in cases of non-electroporated normal intermediate and superficial cells (Nc, black circles, *n* = 645), or electroporated with JMY (Nc _elect_._+JMY_, gray circles, *n* = 463), or electroporation in the absence of JMY (Nc _elect_., green circles, *n* = 99); non-electroporated inflammatory intermediate and superficial cells (Ic, pink circles, *n* = 108), or electroporated with JMY (Ic _elect_._+JMY_, purple circles, *n* = 138). An asterisk means statistically significant differences between groups. Significances were defined as *p* values < 0.05.

**Figure 5 ijms-24-16863-f005:**
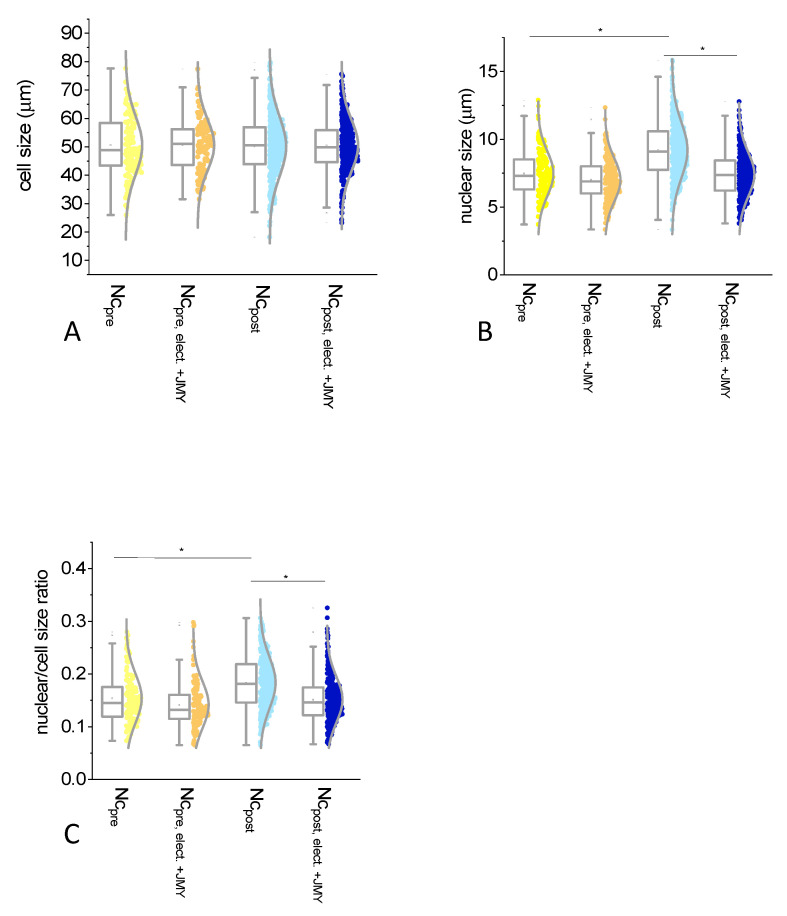
Analysis of the size differences of normal cells, in which cells were non-electroporated or electroporated with JMY from the pre- and post-menopausal subgroups. Box normal diagrams represent data with ±SD (error bars), average value (points), and median values (lines). We studied (**A**) cell size, (**B**) nuclear size, and (**C**) then calculated the nuclear/cell size ratio in cases of non-electroporated normal intermediate and superficial cells from pre-menopausal women (Nc _pre_, yellow circles), or electroporated with JMY (Nc _pre_. _elect_._+JMY_, orange circles); non-electroporated normal intermediate and superficial cells from post-menopausal women (Nc _post_, blue circles), or electroporated with JMY (Nc _post_. _elect_._+JMY_, dark blue circles). Asterisk means statistically significant differences between groups. Significances were defined as *p* values < 0.05.

**Figure 6 ijms-24-16863-f006:**
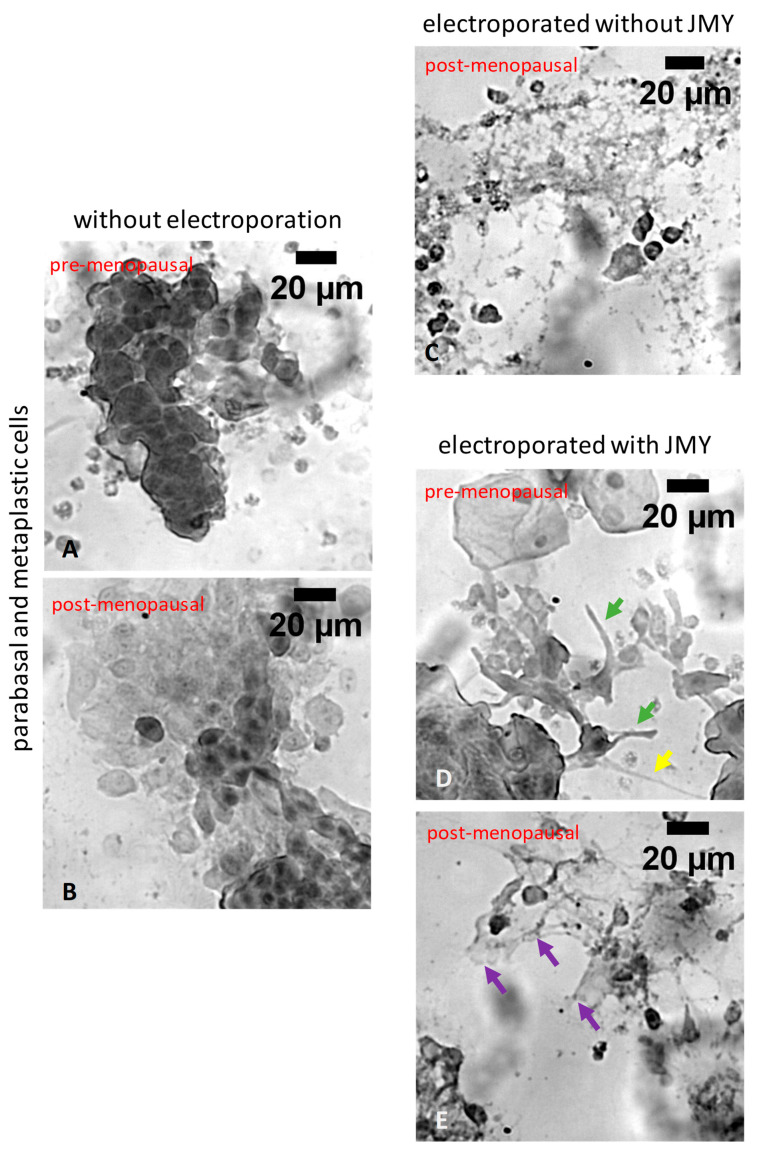
Morphological differences of PAP-stained parabasal and metaplastic cells, in which cells were non-electroporated or electroporated with JMY. (**A**,**B**) Non-electroporated parabasal and metaplastic cells, and (**C**) cells from post-menopausal patients were lysed, only cell debris and nude nuclei remained after the electroporation in the absence of JMY. (**D**) Cells from pre-menopausal samples were reacted with filopodia (green arrow) and membrane tube (yellow arrow) formation after the electroporation with JMY or (**E**) cells from post-menopausal patients in atrophic samples with minor inflammation formed extended cytoplasmic meshwork and protrusions (purple arrows) after the electroporation with JMY.

**Figure 7 ijms-24-16863-f007:**
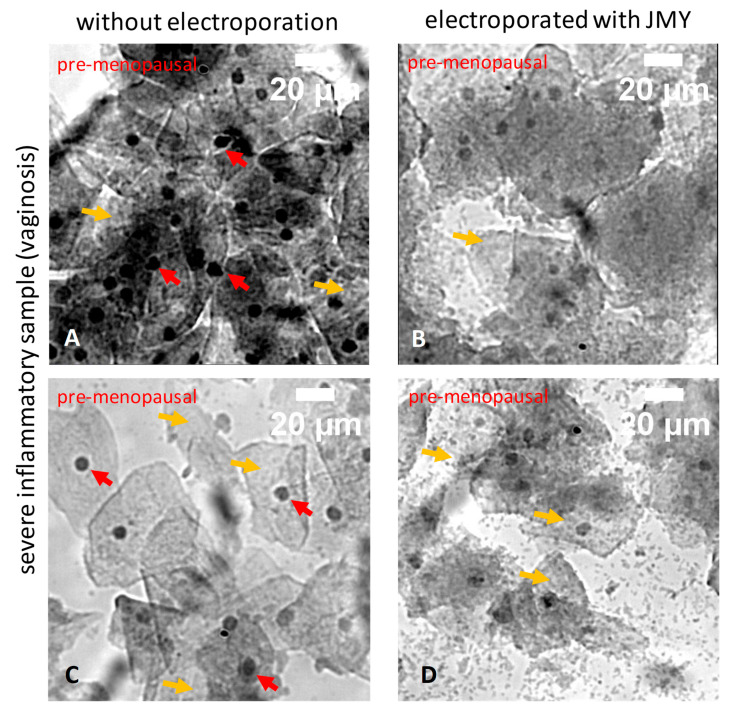
Morphological differences of PAP-stained ectocervical squamous cells from severe bacterial vaginosis in which cells were non-electroporated or electroporated with JMY. (**A**,**C**) Non-electroporated abnormal superficial cells, (**B**,**D**) or abnormal superficial cells electroporated with JMY. Red arrows indicate perinuclear and nuclear irregularities; yellow arrows indicate cytoplasmic clearings.

**Figure 8 ijms-24-16863-f008:**
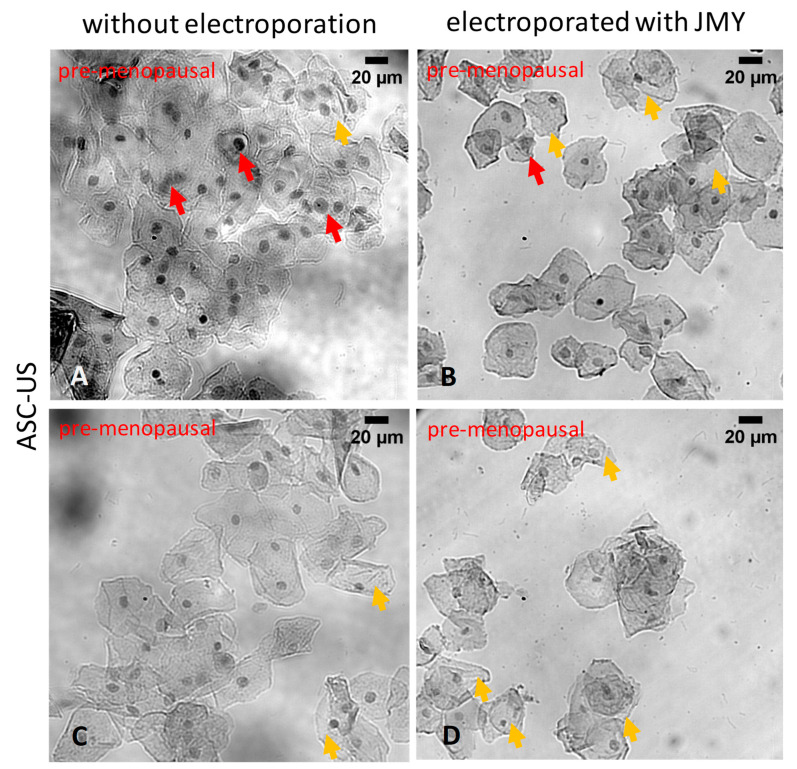
Morphological differences of PAP-stained ectocervical squamous cells from ASC-US sample, in which cells were non-electroporated or electroporated with JMY. (**A**,**C**) Non-electroporated superficial and intermediate cells, (**B**,**D**) or superficial and intermediate cells electroporated with JMY. Red arrows indicate atypical perinuclear and nuclear signs and yellow arrows indicate cytoplasmic clearings.

**Figure 9 ijms-24-16863-f009:**
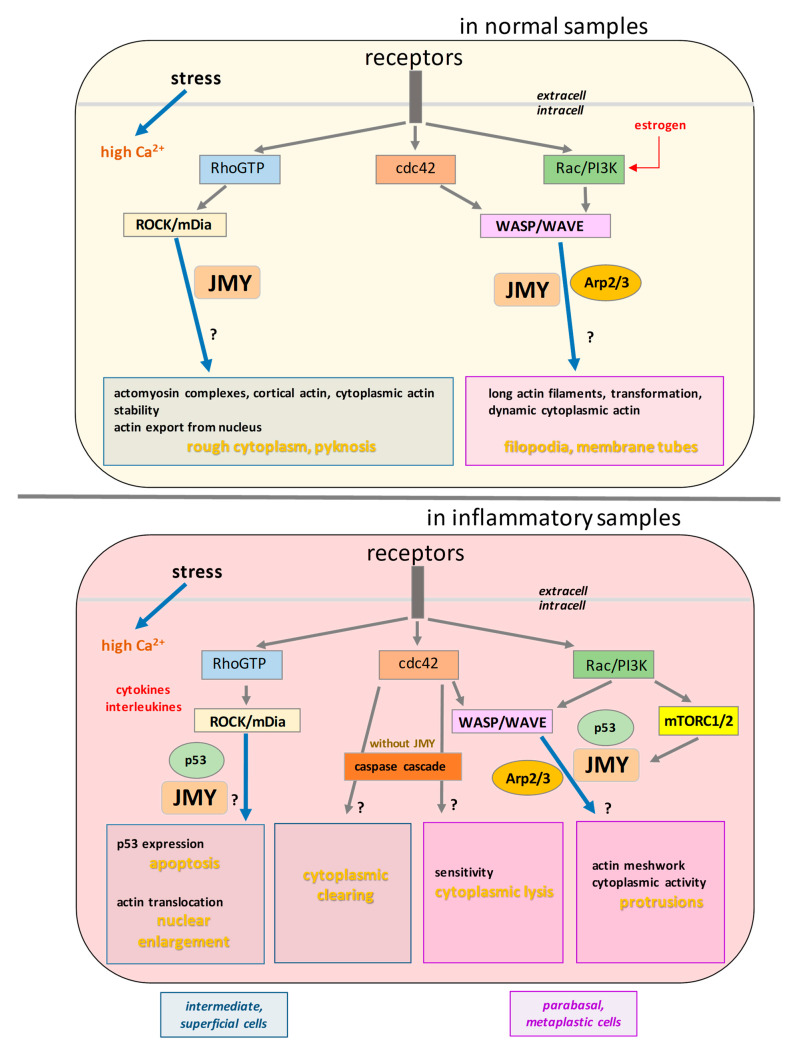
A hypothetical model to interpret the morphological effects of JMY on intracellular pathways. Our observations are indicated with bolded yellow text. (“?” indicates unknown process.)

## Data Availability

The manuscript contains all data which generated within this study.

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
