# Peer review of "Changes of Ex Vivo Cervical Epithelial Cells Due to Electroporation with JMY"

_ijms, 2023, doi:10.3390/ijms242316863_

Round 1
Reviewer 1 Report
Comments and Suggestions for Authors
Overall:
The manuscript titled 'Changes in ex vivo cervical epithelial cells due to JMY electroporation’ examines the morphological changes in ex vivo ectocervical squamous cells after electroporation in the presence of JMY protein. The manuscript's language does not meet the expected standards, rendering it challenging to comprehend and follow. I recommend a thorough revision for clarity, an expanded methods section, and essential English proofreading before considering this manuscript for publication.
More detailed:
Introduction:
The entire introduction appears to be composed in a complex style, making it difficult to follow. Certain sentences seem overly intricate, lacking a clear central idea. Additionally, there's information presented that doesn't appear to directly correlate with the primary focus of the authors' study. Terminology such as 'phylogenetically' and 'JMY protein electroporation' seem to be misused or presented in a context where their relevance is unclear.
There are instances within the manuscript where introduced information lacks the necessary citations. It's imperative to provide appropriate references for all sourced information.
Results:
The term 'cytoplasmic abnormalities' is broad in its current usage. I recommend that the authors provide more specific details and context regarding their observations and statements related to this term.
Discussion:
In this section, the emphasis seems to be primarily on the introduction of the proposed molecular model. This approach somewhat overshadows the actual data on cell/nucleus sizes, leaving it without in-depth discussion. I suggest enhancing the discussion section by comparing the obtained data with previously published results.
4. Materials and Methods
4.1. Proteins
The purification process of the protein appears to be briefly described. Could the authors provide a more detailed account of the procedure or, alternatively, reference a source where the complete methodology is outlined?
4.2. Ex vivo epithelial cells
In the description of how ectocervical squamous cells were acquired, the use of the term 'earned' seems unconventional. Using 'collected' or 'obtained’ to describe this process might be more precise and more standard.
4.3. Low-voltage electroporation to increase the cytoplasmic level of JMY
The electroporation procedure described in this section requires further elaboration. Additionally, details about the steel electrodes, including their measurements and the distance between them, would be beneficial for a comprehensive understanding.
4.4. Microscopy
The section on microscopy requires further clarification: The fixation/staining procedures for the cells aren't described. Could the authors detail the specific methods and materials used for fixation?; It's not specified how much time elapsed between the electroporation treatment and when the cells were fixed. Could this duration be provided?; Was the Olympus IX 81 microscope used in its inverted configuration?
4.5. Image analysis
The section on 'Image analysis' would benefit from a more detailed description to provide clarity on the specific procedures and methodologies employed. Were the cells counted manually?
4.6. Statistics
In the 'Statistics' section, it is mentioned that t-tests were utilized for data comparisons. However, with four or five comparison groups, an Analysis of Variance (ANOVA) might be more appropriate to detect differences among groups. If pairwise t-tests are preferred for specific reasons, have the authors considered applying a correction, such as the Bonferroni correction, to account for multiple comparisons and control the type I error rate?
Figures and captions
Figure 1.
n reference to panels A-D of Figure 1, the label 'JMY electroporated’ is distinctly marked on the right side. However, a corresponding annotation is missing on the left side. Could the authors clarify if the cells in panels A and C were either not electroporated or devoid of JMY introduction? Providing clear labels would ensure transparency and eliminate potential ambiguities. This recommendation is also pertinent for other figures in the manuscript that exhibit similar omissions.
In Panel F, distinctions between pre and post-menopausal women's samples are highlighted. Given these differences, it would be beneficial for clarity if panels A-D also explicitly annotated which samples were being depicted. Ensuring each panel has a clear label regarding the nature of the samples will aid in a thorough understanding of the presented data.
Based on the details provided in section 4.6, a t-test was utilized for the statistical analysis in panels E-F. However, when making comparisons across multiple groups, employing an ANOVA or applying a correction to the t-test may be more appropriate. I would recommend the authors consider one of these approaches to ensure robust statistical analysis.
Figure 2.
Figure 2 seems to lack a structured layout, as the images appear to be misaligned. Ensuring a uniform alignment would enhance clarity and presentation. Moreover, the use of varying colors for annotations (e.g., black, green, and gray) raises questions. If these color choices have specific meanings, they should be clarified in the figure's legend or description (This also applies to other Figures).
Figure 3
See comments for Figures 1-2
Figures 4 and 5
Please refer to the statistical analysis comments provided for Figure 1. Additionally, the specific number of samples included in each group should be clearly stated to enhance clarity and understanding.
Figure 6
See the comments for previous figures. Additionally, I observed black lines in Panels B and C that are not apparent in other panels. Could the authors clarify the origin of these lines?
Figures 7-8
See the comments for previous figures.
Comments on the Quality of English LanguageThe manuscript's language does not meet the expected standards, rendering it challenging to comprehend and follow. Essential English proofreading is needed before considering this manuscript for publication.
Author Response
Summary: We thank the Editorial Board Members and Reviewers for their helpful and constructive comments. Additions or modifications made in direct responses are shown below. Changes made in response to those comments.
Reviewer Comments:
Answers to Reviewer 1.
Overall:
The manuscript titled 'Changes in ex vivo cervical epithelial cells due to JMY electroporation’ examines the morphological changes in ex vivo ectocervical squamous cells after electroporation in the presence of JMY protein. The manuscript's language does not meet the expected standards, rendering it challenging to comprehend and follow. I recommend a thorough revision for clarity, an expanded methods section, and essential English proofreading before considering this manuscript for publication.
Thank you, for these recommendations. We tried to improve the quality of our manuscript.
More detailed:
Introduction:
The entire introduction appears to be composed in a complex style, making it difficult to follow. Certain sentences seem overly intricate, lacking a clear central idea. Additionally, there's information presented that doesn't appear to directly correlate with the primary focus of the authors' study. Terminology such as 'phylogenetically' and 'JMY protein electroporation' seem to be misused or presented in a context where their relevance is unclear.
There are instances within the manuscript where introduced information lacks the necessary citations. It's imperative to provide appropriate references for all sourced information.
Thank you, for these observations. We tried to solve the difficulties with follow of our manuscript. And also, we improved the relevancies and misused phrases, and indicated all relevant references.
Results:
The term 'cytoplasmic abnormalities' is broad in its current usage. I recommend that the authors provide more specific details and context regarding their observations and statements related to this term.
Thank you for this important recommendation, we specified cytoplasmic abnormalities as „cytoplasmic clearing with sharp border” in Line218.
Discussion:
In this section, the emphasis seems to be primarily on the introduction of the proposed molecular model. This approach somewhat overshadows the actual data on cell/nucleus sizes, leaving it without in-depth discussion. I suggest enhancing the discussion section by comparing the obtained data with previously published results.
Thank you for this important suggestion, we completed the rest of the part of Discussion with our observations and tried to interpret them in a comparison with published results, This is the first study which applied JMY transfection in ex vivo cells therefore we can assume the majority of Conclusions.
- Materials and Methods
4.1. Proteins
The purification process of the protein appears to be briefly described. Could the authors provide a more detailed account of the procedure or, alternatively, reference a source where the complete methodology is outlined?
Thank you. We completed it with the relevant reference.
4.2. Ex vivo epithelial cells
In the description of how ectocervical squamous cells were acquired, the use of the term 'earned' seems unconventional. Using 'collected' or 'obtained’ to describe this process might be more precise and more standard.
Thank you for this recommendation.
4.3. Low-voltage electroporation to increase the cytoplasmic level of JMY
The electroporation procedure described in this section requires further elaboration. Additionally, details about the steel electrodes, including their measurements and the distance between them, would be beneficial for a comprehensive understanding.
Thank you. We completed it.
4.4. Microscopy
The section on microscopy requires further clarification: The fixation/staining procedures for the cells aren't described. Could the authors detail the specific methods and materials used for fixation?; It's not specified how much time elapsed between the electroporation treatment and when the cells were fixed. Could this duration be provided?; Was the Olympus IX 81 microscope used in its inverted configuration?
Thank you. We completed it. The preparation of cells was described in the section of „Ex vivo epithelial cells”.
4.5. Image analysis
The section on 'Image analysis' would benefit from a more detailed description to provide clarity on the specific procedures and methodologies employed. Were the cells counted manually?
Thank you. We completed it.
4.6. Statistics
In the 'Statistics' section, it is mentioned that t-tests were utilized for data comparisons. However, with four or five comparison groups, an Analysis of Variance (ANOVA) might be more appropriate to detect differences among groups. If pairwise t-tests are preferred for specific reasons, have the authors considered applying a correction, such as the Bonferroni correction, to account for multiple comparisons and control the type I error rate?
Yes, it is a good idea. ANOVA is a good test if you want to interpret a more generalized variance among populations, and therefore generalizes the t-test beyond two means. Our aim was to obtain and measure the effect of treatment on the population prior to and after the treatment therefore the best parametric test to perform comparisons is if we were using Two sample t-test. The Bonferroni correction is also important in the multiple comparisons.
Figures and captions
Figure 1.
n reference to panels A-D of Figure 1, the label 'JMY electroporated’ is distinctly marked on the right side. However, a corresponding annotation is missing on the left side. Could the authors clarify if the cells in panels A and C were either not electroporated or devoid of JMY introduction? Providing clear labels would ensure transparency and eliminate potential ambiguities. This recommendation is also pertinent for other figures in the manuscript that exhibit similar omissions.
In Panel F, distinctions between pre and post-menopausal women's samples are highlighted. Given these differences, it would be beneficial for clarity if panels A-D also explicitly annotated which samples were being depicted. Ensuring each panel has a clear label regarding the nature of the samples will aid in a thorough understanding of the presented data.
Thank you. We corrected it.
Based on the details provided in section 4.6, a t-test was utilized for the statistical analysis in panels E-F. However, when making comparisons across multiple groups, employing an ANOVA or applying a correction to the t-test may be more appropriate. I would recommend the authors consider one of these approaches to ensure robust statistical analysis.
Yes, it is a good idea. ANOVA is a good test if you want to interpret a more generalized variance among populations, and therefore generalizes the t-test beyond two means. Our aim was to obtain and measure the effect of treatment on the population prior to and after the treatment therefore the best parametric test to perform comparisons if we were using Two sample t-test.
Figure 2.
Figure 2 seems to lack a structured layout, as the images appear to be misaligned. Ensuring a uniform alignment would enhance clarity and presentation. Moreover, the use of varying colors for annotations (e.g., black, green, and gray) raises questions. If these color choices have specific meanings, they should be clarified in the figure's legend or description (This also applies to other Figures).
Thank you. We corrected it.
Figure 3
See comments for Figures 1-2
Figures 4 and 5
Please refer to the statistical analysis comments provided for Figure 1. Additionally, the specific number of samples included in each group should be clearly stated to enhance clarity and understanding.
Thank you. We corrected it.
Figure 6
See the comments for previous figures. Additionally, I observed black lines in Panels B and C that are not apparent in other panels. Could the authors clarify the origin of these lines?
Thank you. We corrected it.
Figures 7-8
See the comments for previous figures.
Comments on the Quality of English Language
The manuscript's language does not meet the expected standards, rendering it challenging to comprehend and follow. Essential English proofreading is needed before considering this manuscript for publication.
Thank you. We tried to improve it.
Thank you for your important and constructive comments, we tried to improve the quality of our manuscript.
Reviewer 2 Report
Comments and Suggestions for Authors
I have had the opportunity to review the manuscript entitled "Changes in ex vivo cervical epithelial cells due to JMY electroporation" submitted to the International Journal of Molecular Sciences (IJMS). The study aims to investigate the effects of JMY electroporation on various cell types, providing a comprehensive analysis of its impact on cell and nuclear size, as well as discussing the potential pathways influenced by JMY treatment.
Major Comments:
Clarity and Organization: The manuscript provides a wealth of data and a comprehensive analysis. However, the presentation of results and the discussion could be made clearer. The authors are encouraged to streamline the content to enhance the readability and ensure that the key findings stand out more prominently.
Hypothetical Model: The authors propose a hypothetical model to explain the observed effects of JMY treatment. While this is a commendable effort, the model could be better visualized. A more detailed and clearer schematic representation of the proposed pathways and mechanisms would significantly aid in the reader’s understanding.
Literature Contextualization: While the manuscript does reference relevant literature, there is room for improvement in how the current findings are contextualized within the existing body of knowledge. The authors should aim to more explicitly highlight how their results contribute to, or differ from, previous studies in the field.
Minor Comments:
Language and Grammar: There are occasional grammatical errors and awkward phrasings throughout the manuscript. A thorough proofreading and language editing would enhance the overall quality of the manuscript.
Reference Formatting: Ensure that all references are correctly formatted and consistent throughout the manuscript.
Conclusion:
The manuscript presents a valuable study with potential implications in understanding the cellular responses to JMY treatment. With the suggested improvements, particularly in terms of clarity, organization, and statistical validation, the manuscript could make a significant contribution to the field. I look forward to seeing the revised version of this manuscript and believe that it has the potential to be a strong addition to the IJMS.
Author Response
Summary: We thank the Editorial Board Members and Reviewers for their helpful and constructive comments. Additions or modifications made in direct responses are shown below. Changes made in response to those comments.
Reviewer Comments:
Answers to Reviewer 2.
I have had the opportunity to review the manuscript entitled "Changes in ex vivo cervical epithelial cells due to JMY electroporation" submitted to the International Journal of Molecular Sciences (IJMS). The study aims to investigate the effects of JMY electroporation on various cell types, providing a comprehensive analysis of its impact on cell and nuclear size, as well as discussing the potential pathways influenced by JMY treatment.
Major Comments:
Clarity and Organization: The manuscript provides a wealth of data and a comprehensive analysis. However, the presentation of results and the discussion could be made clearer. The authors are encouraged to streamline the content to enhance the readability and ensure that the key findings stand out more prominently.
Thank you. We tried to improve it.
Hypothetical Model: The authors propose a hypothetical model to explain the observed effects of JMY treatment. While this is a commendable effort, the model could be better visualized. A more detailed and clearer schematic representation of the proposed pathways and mechanisms would significantly aid in the reader’s understanding.
Thank you. We tried to improve it and modified the text and the cartoon.
Literature Contextualization: While the manuscript does reference relevant literature, there is room for improvement in how the current findings are contextualized within the existing body of knowledge. The authors should aim to more explicitly highlight how their results contribute to, or differ from, previous studies in the field.
Thank you. We tried to improve it.
Minor Comments:
Language and Grammar: There are occasional grammatical errors and awkward phrasings throughout the manuscript. A thorough proofreading and language editing would enhance the overall quality of the manuscript.
Thank you. We tried to improve it.
Reference Formatting: Ensure that all references are correctly formatted and consistent throughout the manuscript.
Thank you. We tried to improve it all References were edited in EndNote X7.
Conclusion:
The manuscript presents a valuable study with potential implications in understanding the cellular responses to JMY treatment. With the suggested improvements, particularly in terms of clarity, organization, and statistical validation, the manuscript could make a significant contribution to the field. I look forward to seeing the revised version of this manuscript and believe that it has the potential to be a strong addition to the IJMS.
Thank you for your important and constructive comments, we tried to improve the quality of our manuscript.

Round 2
Reviewer 1 Report
Comments and Suggestions for Authors
1. In the revised abstract, I noticed that the explanation for the abbreviation 'JMY' is now absent, while other abbreviations like 'CSR' and 'ABP' are clearly defined. For consistency and clarity, I recommend that the authors reintroduce the explanation for 'JMY' in the abstract.
2. There is grammatical error in the abstract - unnecessary period after "chronic inflammation." before "intermediate".
3. Throughout the manuscript, the authors frequently refer to the experimental conditions as 'before' and 'after electroporation'. This phrasing implies that the same cells were observed both before and after the electroporation process. However, considering that the samples for visualization were fixed, it is unlikely that this was the case. To avoid potential misunderstandings, it would be more accurate for the authors to redefine the experimental groups. The suggested terminology could be 'non-electroporated (sham-exposed)' and 'electroporated' samples. This change would more precisely reflect the experimental design and enhance the clarity of the results presented
4. Statistics. The authors state that their comparison was limited to non-electroporated versus electroporated with JMY cells. However, this does not accurately reflect the entirety of the comparisons conducted. The study also includes comparisons between non-electroporated cells and electroporated cells. Additionally, the data presented in some figures indicate significant differences between pre-menopausal and post-menopausal samples. This suggests that multiple t-tests were utilized to determine significance. It is important to note that the use of multiple t-tests in this manner may lead to a reduction in the statistical power of the analysis. This aspect warrants a more thorough consideration to ensure the validity and reliability of the study's findings.
5. METHODS. Section "Low-voltage electroporation to increase the intracellular level of JMY"
A more detailed description of the electroporation setup utilized in the study is required. Specifically, it would be informative to know whether the cells were in suspension or attached to the culture dish at the time of electroporation. Additionally, the volume of the medium in which the cells were contained during this process should be clarified. Moreover, the authors mention that an electric field of 5 V/cm was employed. It would be helpful to specify where exactly this electric field strength was expected to be achieved — across the entire volume of the medium (or electroporated area), locally around the electrodes, or in a different configuration. Such details are crucial for fully understanding the experimental conditions and replicating the study in future research.
Comments on the Quality of English LanguageSome sentences are quite long and complex. Breaking them into shorter sentences might improve readability.
Author Response
Summary: We thank the Editorial Board Members and Reviewers for their helpful and constructive comments. Additions or modifications made in direct responses are shown below. Changes made in response to those comments.
Reviewer Comments:
Answers to Reviewer 1.
- In the revised abstract, I noticed that the explanation for the abbreviation 'JMY' is now absent, while other abbreviations like 'CSR' and 'ABP' are clearly defined. For consistency and clarity, I recommend that the authors reintroduce the explanation for 'JMY' in the abstract.
Thank you, for this recommendation. We tried to improve the quality of our manuscript. We explained JMY in the part of the Abstract in Line16.
- There is grammatical error in the abstract - unnecessary period after "chronic inflammation." before "intermediate".
Thank you, for this observation. We corrected it in Line31.
- Throughout the manuscript, the authors frequently refer to the experimental conditions as 'before' and 'after electroporation'. This phrasing implies that the same cells were observed both before and after the electroporation process. However, considering that the samples for visualization were fixed, it is unlikely that this was the case. To avoid potential misunderstandings, it would be more accurate for the authors to redefine the experimental groups. The suggested terminology could be 'non-electroporated (sham-exposed)' and 'electroporated' samples. This change would more precisely reflect the experimental design and enhance the clarity of the results presented
Thank you, for this important recommendation. We corrected them in Lines170, 175, 195, 203, 216, 221, 240, 258, 271, 279.
- The authors state that their comparison was limited to non-electroporated versus electroporated with JMY cells. However, this does not accurately reflect the entirety of the comparisons conducted. The study also includes comparisons between non-electroporated cells and electroporated cells. Additionally, the data presented in some figures indicate significant differences between pre-menopausal and post-menopausal samples. This suggests that multiple t-tests were utilized to determine significance. It is important to note that the use of multiple t-tests in this manner may lead to a reduction in the statistical power of the analysis. This aspect warrants a more thorough consideration to ensure the validity and reliability of the study's findings.
Thank you, for this important interpretation. We tested all analysed data with ANOVA and Bonferroni tests (see Supplemented Tables) as you recommended previously and we modified Fig.4A, 5B,C and Line208 as a response to the statistical significance in view of variance of data. And corrected in part of Materials and Methods in Line453. We can suggest that the ANOVA and Bonferroni analysis resulted in minor changes to our previous conclusions but improved the validity and reliability of this manuscript.
- METHODS. Section "Low-voltage electroporation to increase the intracellular level of JMY"
A more detailed description of the electroporation setup utilized in the study is required. Specifically, it would be informative to know whether the cells were in suspension or attached to the culture dish at the time of electroporation. Additionally, the volume of the medium in which the cells were contained during this process should be clarified. Moreover, the authors mention that an electric field of 5 V/cm was employed. It would be helpful to specify where exactly this electric field strength was expected to be achieved — across the entire volume of the medium (or electroporated area), locally around the electrodes, or in a different configuration. Such details are crucial for fully understanding the experimental conditions and replicating the study in future research.
Thank you, for this important recommendation. We corrected them in Lines427, 431, 436.
Comments on the Quality of English Language
Some sentences are quite long and complex. Breaking them into shorter sentences might improve readability.
Thank you, for this important suggestion. We corrected them in Lines19, 50, 56, 325, 342, 359, 371.
Thank you for your important and constructive comments, we tried to improve the quality of our manuscript.